# Gefitinib-Induced Severe Dermatological Adverse Reactions: A Case Report and Pharmacogenetic Profile

**DOI:** 10.3390/ph17081040

**Published:** 2024-08-07

**Authors:** Mariana Vieira Morau, Cecilia Souto Seguin, Mauricio Wesley Perroud Junior, Carolina Dagli-Hernandez, Eder de Carvalho Pincinato, Patricia Moriel

**Affiliations:** 1Faculdade de Ciências Médicas, Universidade Estadual de Campinas, Campinas 13083-887, SP, Brazil; marianavmorau@gmail.com (M.V.M.); ceciliaseguin@gmail.com (C.S.S.); mperroud@unicamp.br (M.W.P.J.); edercp@unicamp.br (E.d.C.P.); 2Faculdade de Ciências Farmacêuticas, Universidade Estadual de Campinas, Campinas 13083-871, SP, Brazil; carolina.dagli@gmail.com

**Keywords:** gefitinib, adverse drug reaction, skin rash, pharmacogenetics, *ABCB1*, cytochrome P450 enzyme system, case report

## Abstract

Gefitinib is a selective inhibitor of the epidermal growth factor receptor that is used to treat advanced and metastatic non-small cell lung cancer (NSCLC). Dermatological adverse reactions are most commonly associated with gefitinib treatment. The cause of adverse reactions in individuals is multifactorial. Pharmacogenetics is an effective tool to detect such adverse reactions. This case report describes a female patient with NSCLC who was administered gefitinib at a dose of 250 mg/day. However, due to severe adverse dermatological reactions, the treatment was interrupted for 15 d and antibiotic therapy was administered to manage the skin rashes, maculopapular rashes, and hyperpigmentation. Treatment adherence was adequate, and no drug interactions were detected. A pharmacogenetic analysis revealed homozygosity in the ATP-binding cassette *(ABC)-B1* rs1128503 (c.1236A>G), heterozygosity in *ABCG2* rs2231142 (c.421G>T) and rs2622604 (c.-20+614T>C), and a non-functional variant of the cytochrome P450 family 3, subfamily A, member 5 (*CYP3A5*). The relationship between altered genetic variants and the presence of adverse reactions induced by gefitinib is still controversial. Overall, this case report highlights the importance of continuing to study pharmacogenetics as predictors of adverse drug reactions.

## 1. Introduction

Gefitinib is a potent oral and selective epidermal growth factor receptor (EGFR) tyrosine kinase inhibitor (TKI) [1]. It is an effective therapeutic agent for non-small cell lung cancer (NSCLC) associated with active mutations in EGFR, mainly in exons 19 (deletion) and 21 (L858R) [2]. TKI-EGFR is commonly associated with dermatological adverse events (AEs; observed in almost 50% of the patients), as EGFR is expressed in the basal layers of epidermal keratinocytes and hair follicles, where it plays essential roles in cell structure maintenance and proliferation [3,4].

The main dermatological clinical manifestations include rashes, maculopapular rashes, and paronychia. These AEs occur due to EGFR blockade, which leads to endothelial inflammation, reduced vascular tone, and consequently, increased vascular permeability [5]. Therefore, dermatological manifestations are secondary adverse drug reactions, that is, with predictable and non-immunological effects, thus differentiating themselves from allergic and hypersensitivity reactions [6,7].

Gefitinib is extensively metabolized in the liver, predominantly by the cytochrome P450 (CYP) enzyme, *CYP3A4*, and moderately by *CYP3A5* and *CYP2D6*. Its transport out of the cells is dependent on ATP-binding cassette (ABC) transporters, such as *ABCG2* (BCRP/MRP) and *ABCB1* (P-glycoproteins/MRD1) [8,9]. Pharmacogenetic studies on the single-nucleotide polymorphisms (SNPs) of CYP enzymes and ABC transporters have revealed the important pharmacokinetic and pharmacodynamic differences between them, possibly responsible for the occurrence of AEs [10,11,12,13]. Human leukocyte antigen (HLA)-A and -B have also been analyzed to assess the possible influence of *HLA* genotypes on the response to antineoplastic treatments [14].

Here, we present a case report of a female patient with NSCLC who experienced severe adverse dermatological events after gefitinib treatment. We investigated whether the patient carried any variants in the genes involved in the pharmacogenetics of gefitinib and analyzed the drug–drug interactions that could have led to the observed AEs.

This study is reported in accordance with the CARE Guidelines and Checklist available on the CARE website (www.care-statement.org) and EQUATOR Network (www.equator-network.org) (accessed on 10 May 2024).

Written informed consent was obtained from the patient for the publication of this case report and any accompanying images.

## 2. Results

A 55-year-old Caucasian woman was admitted to the OncoPneumology Ambulatory of a public hospital at UNICAMP (Campinas, São Paulo, Brazil) in March 2022. She was diagnosed with adenocarcinoma of the right lung with metastasis to the mediastinal lymph nodes, liver, and bone. Genetic tests for *EGFR* showed that the patient had a mutation in L858R in exon 21, after which she was prescribed gefitinib at 250 mg/day.

The patient did not smoke at the time but had been a passive smoker for approximately 10 years. No alcoholism or other comorbidities were reported. The patient did not have a history of hypersensitivity reactions to drugs. Other medications used at the time of gefitinib prescription included metamizole (500 mg) for pain control, doxycycline twice daily, and estriol ointment prescribed by an outside gynecologist for a urinary infection.

After 4 weeks of gefitinib treatment, the patient developed an acneiform rash on the face and scalp, with hyperpigmentation on the face (Figure 1a). Topical clindamycin and hydrocortisone were prescribed for AEs, and gefitinib treatment was not interrupted.

Adverse dermatological reactions worsened significantly after 10 weeks of gefitinib treatment. The patient exhibited pustules and papules on her face and maculopapular and acneiform eruptions, which was characterized as grade 3 according to the Common Terminology Criteria for Adverse Events (CTCAE) [15] (Figure 1b,c). At this point, treatment with gefitinib was discontinued for 15 days, and antibiotic therapy with tetracycline (1 g) was initiated twice daily. As shown in Figure 1d–f, a slight improvement in adverse reactions was observed 7 d after gefitinib interruption and initiation of oral tetracycline therapy.

After interruption of the targeted therapy, the patient returned to the ambulatory state with significant improvement in skin lesions (Figure 1g–i). Therefore, gefitinib was reintroduced (250 mg/day), and tetracycline treatment was continued at 500 mg twice daily.

In November 2022, 8 months after starting gefitinib treatment, the patient reported improvement in rashes on her face. However, acneiform rashes appeared on the abdomen and right and left sides of the costal region (Figure 1j–m).

To identify the possible predisposing genetic polymorphisms contributing to the etiology of these serious adverse reactions, the following genes encoding enzymes and transporters involved in the pharmacokinetics and pharmacodynamics of gefitinib were investigated: *EGFR*, *ABCB1*, *ABCG2*, *CYP2D6*, *CYP3A4*, *CYP3A5*, and *HLA*. Analyses were performed by microarray using the Infinium Global Diversity Array with Enhanced PGx (iSan Illumina, San Diego, CA, USA).

The patient reported adherence to treatment. According to the Brief Medication Questionnaire (BMQ) and MEDTAKE [16,17,18], the patient did not forget her medications (100% adherence).

The results of the genetic tests are shown in Table 1. In drug transporters, the patient is homozygous for *ABCB1* rs1128503 (c.1236A>G) and heterozygous for *ABCG2* rs2231142 (c.421G>T) and rs2622604 (c.-20+614T>C). The genotypes of the CYP enzymes were *CYP2D6 *1/*2* and *CYP3A5 *3/*3* (rs776746, c.219-237T>C). She did not carry any variants in CYP3A4 or EGFR, nor did she carry any variants in *HLA* types.

Next, we investigated whether drug–drug interactions cause AEs. We identified doxycycline as a possible inhibitor of *CYP3A4* [19], which is the main enzyme metabolizing gefitinib.

## 3. Discussion

EGFR inhibitors are often used as monotherapies or in combination with radiotherapy for the treatment of solid tumors, such as lung, colorectal, and pancreatic cancers [20]. NSCLC is the most common subtype of lung cancer, accounting for 80–85% of all cases [21]. Gefitinib is a first-generation selective EGFR inhibitor used for patients with advanced and metastatic NSCLC; it is well tolerated orally, but some adverse reactions have been reported [22].

Most adverse reactions associated with the use of gefitinib are linked to the integumentary, gastrointestinal, and hepatic systems, the vast majority of which are mild to moderate in nature (grade 1–2 according to the CTCAE) [15,23]. Dermatological AEs, especially acneiform and maculopapular rashes (mainly in the face, chest, neck, and back; 47% of patients) and dry skin (13%), are among the most common adverse reactions (50–55% of patients) [21,22,23,24]. Some studies have also reported adverse nail reactions, such as paronychia and onychocryptosis [9,25,26], and a few have reported alopecia [27,28].

Some studies have reported dermatological adverse reactions, such as severe and extensive skin rashes with redness and aspparent swelling requiring the discontinuation of gefitinib [29] as well as rashes with scaly crusts on the face requiring the cessation of TKIs [30]. However, to date, no study has investigated the association of pharmacogenetics with the occurrence of adverse reactions.

Here, the patient was a homozygous carrier of the *CYP3A5*3* (rs776746, c.219-237T>C) variant, which results in enzyme inactivity [31]. A study in healthy individuals showed that this polymorphism, once detected, directly alters the pharmacokinetics of gefitinib, specifically its clearance [32]. On the other hand, a study in individuals with NSCLC suggested that this polymorphism in *CYP3A5* would not be associated with adverse reactions induced by gefitinib, much less with plasma levels of the drug. However, the *CYP3A4*1/*1G* variant, for which the patient was not a carrier, is associated with this risk of toxicity [33]. *CYP3A5* and *CYP3A4* share 83% similarity in their amino acid sequences, possessing significant structural homology [31].

Gefitinib metabolism is quite extensive and complex [34]. An in vitro study comparing TKIs showed that gefitinib is more susceptible to metabolism by CYP3A5, which may contribute to higher clearance [10]. However, another in vitro study with human liver microsomes inferred that CYP3A5 has much lower catalytic activity (about 12%) towards gefitinib compared to CYP3A4 [35]. To date, only one study in patients with NSCLC has shown that a poor metabolizer profile in CYP3A5 can predict ADR, particularly hepatotoxicity [36]. CYP3A5*3 impairs the elimination of gefitinib; nevertheless, there is controversy over whether it can cause adverse reactions induced by gefitinib.

*ABCB1* or MDR1 encodes the P-glycoprotein (P-gp) region containing approximately 124 SNPs [37] that have been abundantly studied owing to their important roles in drug efflux [38]. This protein is expressed in the liver, kidneys, and lungs [39]. Here, the patient carried the GG genotype of *ABCB1* rs1128503 and presented with severe skin reactions. Preemptive genotyping of some *ABCB1* SNPs, rs1128503 (c.1236A>G) in this case, can help to reduce the adverse reactions to chemotherapy drugs [40]. In colorectal cancer, capecitabine-treated patients with GG genotype exhibit a higher risk of neutropenia or hand–foot syndrome compared to those with AA genotype [41]. However, some studies on gefitinib have reported otherwise. A study of Chinese patients with NSCLC revealed that the GG genotype reduces the risk of diarrhea and skin rashes on gefitinib treatment [42]. In contrast, a systematic review with meta-analysis carried out by our group [43] demonstrated possible clinical implications of *ABCB1* rs1128503 (c.1236A>G) in gefitinib-induced rash and diarrhea reactions. Therefore, the skin reactions observed in our patient may have been caused by this variant.

*ABCG2* is an important gene involved in the efflux of several drugs, including gefitinib [44]. Here, the patient was heterozygous for both *ABCG2* rs223142 (421G>T) and rs2622604 (c.-20+614C>T). The consequences of these polymorphisms have not yet been fully elucidated, but one study showed that individuals carrying the GT genotype of the 421G>T (rs223142) polymorphism exhibit an increased risk of diarrhea compared to those with the GG genotype after gefitinib treatment [45]. However, a study with patients with NSCLC did not find associations with the T allele (heterozygous or homozygous) and gefitinib reactions [46]. The presence of adverse reactions, such as skin reactions or diarrhea, can be explained by the roles of transporters in the oral absorption and elimination pathways of gefitinib, as *ABCG2* is highly expressed in the intestine and liver [47]. A meta-analysis study that investigated the associations of *ABCG2* genes and gefitinib toxicities showed that rs rs223142 (421G>T) may not be a reliable biomarker of dermatological toxicity induced by gefitinib [48]. Thus, the association of *ABCG2* rs223142 in the current patient is questionable and warrants further investigation with additional cases.

*CYP* genes are highly polymorphic. Failure or low activity of CYP enzymes [49] and genetic copy number variations (CNVs) [50] increase the risk of adverse reactions. Reduced *CYP2D6* function (*10/*10) is associated with an increased risk of grade ≥2 skin rashes [43,51]; however, patients with *CYP2D6* metabolizing genes exhibit a normal genotype, *CYP2D6*1/*2*, with a CNV of 2.

*HLA* genes, despite not being closely related to the metabolism of gefitinib, were investigated in this study, as they have high germinal and somatic heterogeneity and influence the therapy of patients’ cancer [14]. Patients with cancer who are homozygous for *HLA* class I exhibit reduced responses to checkpoint inhibitors [52]. However, in this study, the patient exhibited a normal *HLA* genotype.

Whether non-genetic factors, such as drug–drug interactions, lead to adverse reactions warrants further investigation. Here, the patient was treated with doxycycline, metamizole, and estriol ointments when the gefitinib treatment was initiated. An in vitro study demonstrated that doxycycline induces *CYP3A4* [19]. As gefitinib is also metabolized by *CYP3A4*, this could be a possible mechanism of drug–drug interaction. However, no other studies support this finding, and no evidence of the interactions between doxycycline and gefitinib is currently available. Therefore, we excluded the possibility of drug–drug interactions in this study.

This study has some limitations. As this is a case report, it does not indicate a representative sample of the population; however, it highlights the importance of pharmacogenetic studies in predicting adverse drug reactions. The plasma concentration of gefitinib in this patient was within the expected therapeutic window, as it was orally administered. Moreover, the patient adhered to the medication plan.

## 4. Conclusions

Here, our findings showed a case of a patient with NSCLC, using gefitinib, doxycycline, and metamizole, plus homozygosity in *ABCB1* rs1128503 (c.1236A>G), heterozygosity in *ABCG2* rs2231142 and rs2622604 (c.421G>T+c.-20+614T>C), and the presence of a non-functional component in the *CYP3A5* variant, and the presence of moderate to severe adverse reactions associated with the use of gefitinib.

There are several controversies about the contribution of these variants in the adverse reactions presented by the patient. However, we highlight the hypothesis that the ADRs induced by gefitinib may be caused by the non-functional variant of *CYP3A5*3* and in the homozygous variant of *ABCB1* (rs1128503). We emphasize the importance of continuous study of pharmacogenetic analyses in predicting individual responses to treatment, particularly concerning adverse drug reactions. This is essential for advancing research on genetic variants as potential risk biomarkers.

## Figures and Tables

**Figure 1 pharmaceuticals-17-01040-f001:**
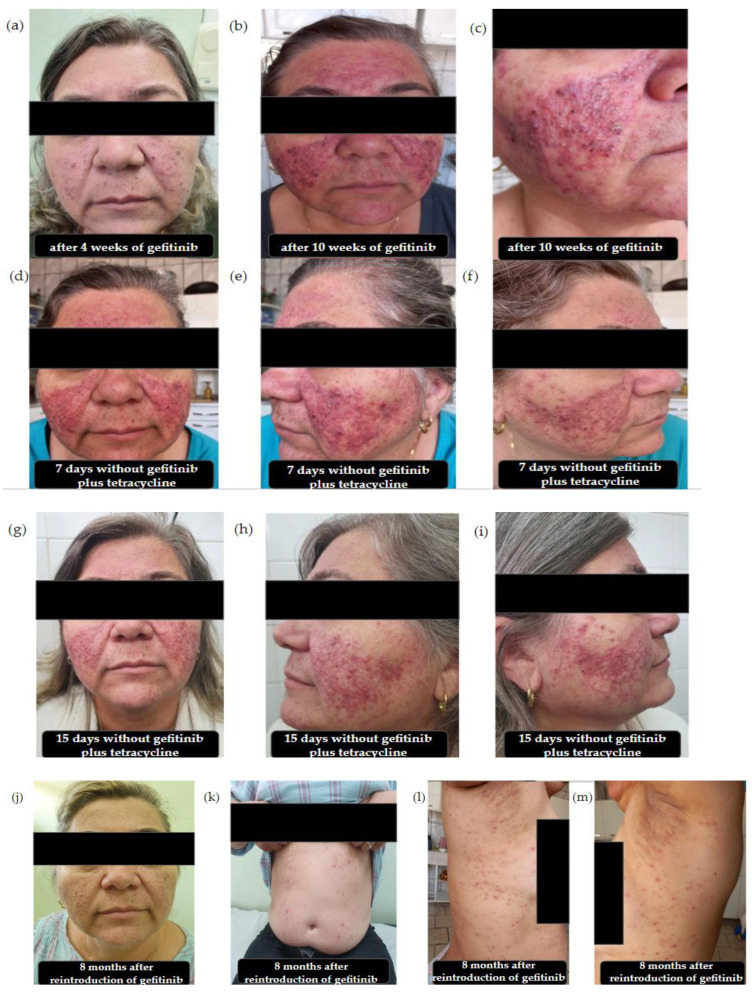
Adverse reactions induced by gefitinib (250 mg/day) in a 55-year-old woman. (**a**) Mild cutaneous adverse reactions on the face after 4 weeks of gefitinib treatment. (**b**) Severe cutaneous adverse reactions on the face after 10 weeks of gefitinib treatment. (**c**) Appearance of pustules and papules on the left side of the cheek after 10 weeks of gefitinib treatment. (**d**) Cutaneous adverse reactions on the face 7 d after interrupting gefitinib treatment and starting oral tetracycline (1 g twice daily) treatment. (**e**) Left side of the face 7 d after interrupting gefitinib treatment and starting oral tetracycline (1 g twice daily) treatment. (**f**) Right side of the face 7 d after interrupting gefitinib treatment and starting oral tetracycline (1 g twice daily) treatment. (**g**) Cutaneous adverse reactions on the face 15 d after interrupting gefitinib treatment and starting oral tetracycline treatment. (**h**) Left side of the face 15 d after interrupting gefitinib treatment and starting oral tetracycline treatment. (**i**) Right side of the face 15 d after interrupting gefitinib treatment and starting oral tetracycline treatment. (**j**) The face showed a serious adverse reaction 8 months after the re-introduction of gefitinib. (**k**) Patient abdomen exhibited mild adverse reactions 8 months after the re-introduction of gefitinib. (**l**) Right side of the costal region exhibited adverse reactions after the re-introduction of gefitinib. (**m**) Left side of the costal region exhibited adverse reactions after the re-introduction of gefitinib.

**Table 1 pharmaceuticals-17-01040-t001:** Pharmacogenetic tests results.

Gene	Patient Genotype	Patient Phenotype
*CYP2D6*	*1/*2	normal metabolizer (CNV = 2)
*CYP3A4*	*1/*2	normal metabolizer
*CYP3A5*	*3/*3	poor metabolizer
*ABCB1*		
rs 1045642 (3435 A > C)	A/A	-
rs 1128503 (c.1236A > G)	G/G	-
rs 2032582 (2677 C > T/A)	C/C	-
*ABCG2*		
rs2231142 (c.421G > T)	G/T	-
rs2622604 (c.-20 + 614 T > C)	C/T	-
*EGFR*		
rs2227983 (1562 G > A)	G/G	-
rs2293347 (c.2982 C > T)	C/C	-
*HLA*		
A/B	Ref/Ref	-

*CYP*, cytochrome P450; *ABCB1*, adenosine triphosphate-binding cassette subfamily B member 1; *ABCG2*, adenosine triphosphate-binding cassette subfamily G member 2; *EGFR*, epidermal growth factor receptor; *HLA*, human leukocyte antigen; CNV, copy number variation.

## Data Availability

The data presented in this study are openly available in Patricia Moriel, 2024, “Replication Data for: Influência das variantes genéticas (SNPs) e da concentração plasmática de gefitinibe nas reações adversas e sobrevida de pacientes com câncer de pulmão de não pequenas células”, https://doi.org/10.25824/redu/DPBPXU, Repositório de Dados de Pesquisa da Unicamp.

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
