# Peer review of "Gefitinib-Induced Severe Dermatological Adverse Reactions: A Case Report and Pharmacogenetic Profile"

_pharmaceuticals, 2024, doi:10.3390/ph17081040_

Round 1
Reviewer 1 Report
Comments and Suggestions for Authors
Dear Authors
Thank you for your manuscript.
I have minor comments.
For clinical practice, do the authors recommend to check pharmacogenetic in all patients who are going to start EGFR inhibitors. As those dermatological reactions are adverse events, it is not drug allergy. So those side effects are expected to occur in all patients with EGFR inhibitors. Although the pharmacogenetic is positive, the level of EGFR inhibitor is not in therapeutic window. Thus, it is also difficult to use it for adjust drug dosage.
How this pharmacogenetic can be useful for clinical practice?
Best wishes
Author Response
Title of the manuscript: Gefitinib-Induced Severe Dermatological Adverse Reactions: A Case Report and Pharmacogenetic Profile
Dear Editor and Reviewers,
We very much appreciated the considerations about our work, which helped us to improve the manuscript. We reviewed the suggested changes in the text and answered the questions. Please, find below your comments addressed. To make it easier for you to read, we leave all changes in the manuscript highlighted by using the "Track Changes" function in Microsoft Word, or highlighted in yellow, or when more appropriate, we leave a comment explaining the change. Please, do not hesitate in contacting us if there are any issues left behind or new comments and suggestions.
Kind regards,
Reviewer 1
For clinical practice, do the authors recommend to check pharmacogenetic in all patients who are going to start EGFR inhibitors. As those dermatological reactions are adverse events, it is not drug allergy. So those side effects are expected to occur in all patients with EGFR inhibitors. Although the pharmacogenetic is positive, the level of EGFR inhibitor is not in therapeutic window. Thus, it is also difficult to use it for adjust drug dosage.
How this pharmacogenetic can be useful for clinical practice?
Author's reply: We thank the reviewer for the valuable comments to our paper. We clarify that our intention was not to make a recommendation based solely on this case. We cannot and should not extrapolate the data. However, with this report, our goal is to highlight the significant influence that pharmacogenetics and personalized medicine have on the quality of life of a patient using TKIs, where the percentage of dermatological adverse reactions is nearly 90% (Li, Yanping et al, 2022). Therefore, preemptive pharmacogenetic testing in the future may help to identify which patients using TKI are at higher risk of presenting dermatological adverse reactions.
In attention to your comments, we have added a paragraph to the manuscript, which can be read below, clarifying the difference between adverse drug reactions and allergic reactions, and that these will occur regardless of whether the plasma concentration is within the therapeutic window for the drug in question (Coleman JJ, et al, 2016. PMID: 27697815; FO Ajayi, et al 2000. PMID: 11028248)
Inserted paragraph in introduction (1)
Lines (38-43)
The main dermatological clinical manifestations include rashes, maculopapular rashes and paronychia. These AEs occur due to EGFR blockade, which leads to endothelial inflammation, reduced vascular tone and, consequently, increased vascular permeability [5]. Therefore, dermatological manifestations are secondary adverse drug reactions, that is, with predictable and non-immunological effects, thus differentiating themselves from allergic and hypersensitivity reactions [6], [7] .

Reviewer 2 Report
Comments and Suggestions for Authors
Author Response
This case report describes adverse reactions in a Non-Small-Cell Lung cancer (NSCLC) patient being treated with Gefitinib, metamizole, doxycycline and estriol ointment. The authors performed an extensive pharmacogenomic analysis using Illumina’s global diversity array which examines some 6,000+ variants and found six variant genotypes of potential association with gefitinib (Table 1). Of these, three are for the CYP enzymes (one each for CYP2D6, CYP3A4 and CYP3A5) one for ABCB1 transporter (rs1128503; GG homozygous; the other two genotypes listed for ABCB1 in Table 1 are NOT variants) and two for ABCG2 transporter (heterozygous for rs2231142 as well as s2622604).
First, variants for CYP2D6 and CYP3A4 (both enzymes with a major role in gefitinib metabolism) have a normal phenotype, thus unlikely to be associated with the adverse reactions in the present case. The CYP3A5 poor metabolizer is a predominant variant across the globe, also indicating that it is unlikely but possible that CYP3A5*3/*3 genotype may be contributing to the adverse reactions in this one case. Hence, I would give the authors a benefit of doubt in this case.
Second, variants in the transporter genes have been studied extensively in connection with gefitinib therapy. Of these, the homozygous ABCB1 variant rs1128503 (GG) has been shown to have no effect on gefitinib metabolism (Hisrose Takasi et al. 2016; PMID 26898617). The authors also list several references and argue how this variant may NOT be the cause of adverse reactions observed in the present case (Lines 161-173) and yet suggest, under ‘Conclusions’, that ‘homozygosity in ABCB1 rs 1128503 is responsible for the adverse reactions associated with gefitinib.’ Therefore, this conclusion cannot be accepted.
Association of both ABCG2 variants found in this case study have also been examined extensively. As listed by the authors in references 40 (Cusatis et al. ;2006), variant GT for rs223142 showed increased risk for diarrhea after gefitinib treatment, which the authors extrapolate to skin-related adverse reactions in the present study, possibly due to increased absorption of gefitinib in this patient. However, Akasaka Keiichi et al. (2010; PMID 20035425), in a study involving 75 NSCLC patients found no association of allele T, heterozygous or homozygous form, with adverse effects of gefitinib. Moreover, reference 38 in the manuscript is another more recent study by Kobayashi Hiroyuki et al. (2015; PMID 22554506) in NSCLC patients showed no difference in concentrations of gefitinib in patients with genotypes GG, GT or TT, questioning the proposed mechanism in the present study (Figure 1) with a single patient.
Genotypes for two other genes listed in Table 1 (EGFR and HLA A/B) appear to be normal and have not been implicated as being associated with the observed adverse reactions in this patient.
In summary, the tenuous pharmacogenetic associations with the observed adverse reactions to gefitinib in this study are questionable at best. Therefore, I am unable to recommend this manuscript for publication in the present form. Having said that, the mild to severe adverse reactions the patient has experienced cannot be ignored. The significant effort that the authors have put forth in conducting pharmacogenetic analysis is also praiseworthy and should be recognized. Therefore, I am suggesting that the authors revise their conclusions in light of the above cited references. The well-documented contribution of doxycycline to the dermatological adverse reactions should be included under ‘Conclusion’. Contributions, if any, of metamizole, also cannot be ruled out.
Author's reply: We thank the reviewer for their thorough review of our paper and for the observations and questions. We will answer them below by gene and highlight the excerpts added to the manuscript in yellow.
CYP3A5 gene: as our manuscript mentions a divergence regarding the inactivity of this CYP3A5 enzyme and adverse reactions, but to corroborate the matter, we brought the article by Kwok WC, 2022 and collaborators, who demonstrated in a cohort of patients with NSCLC that CYP3A5 polymorphisms were not associated with adverse reactions induced by gefitinib.
Rewritten fourth paragraph of the discussion (3)
(lines 156-165)
Here, the patient was a homozygous carrier of the CYP3A5*3 (rs776746, c.219-237T>C) variant, which results in enzyme inactivity [32]. A study in healthy individuals showed that this polymorphism, once detected, directly alters the pharmacokinetics of gefitinib, specifically its clearance [33]. On the other hand, a study in individuals with NSCLC suggested that this polymorphism in CYP3A5 would not be associated with adverse reactions induced by gefitinib, much less with plasma levels of the drug. However, the CYP3A4*1/*1G variant, for which the patient was not a carrier, is associated with this risk of toxicity [34]. CYP3A5 and CYP3A4 share 83% similarity in their amino acid sequences, with overlapping substrate specificities [32]. CYP3A5*3 impairs the elimination of gefitinib, nevertheless, there is controversy over whether it can cause adverse reactions induced by gefitinib.
ABCB1 gene: there are diverging results about the association of the ABCB1 rs1128503 (GG) polymorphic variant with adverse reactions induced by gefitinib. As mentioned by the reviewer, this fact is addressed in the paragraph referring to the gene (lines 161-173). In the study by Hirose T. and collaborators, the authors did not directly investigate the association between polymorphisms and the degrees of adverse reactions observed with the use of gefitinib. The focus of the study was on the relationship between plasma concentration and genes, including ABCB1 rs1128503, and no positive association was found between pharmacokinetics and pharmacogenomics. To support this hypothesis that rs1128503 may be involved in adverse reactions, we present a systematic review and meta-analysis conducted by our group, [Morau MV et al.,2024]. This review included 5 studies with populations of 3 different ethnicities. Eight genetic variants of ABCB1 were studied, highlighting the rs1045642 (3435C>T) and rs1128503 (1236C>T) variants, which demonstrated possible clinical implications in skin rash and diarrhea reactions in patients with NSCLC.
Rewritten fifth paragraph of the discussion (3)
(lines 167-180)
ABCB1 or MDR1 encodes the P-glycoprotein (P-gp) region containing approximately 124 SNPs [35] that have been abundantly studied owing to their important roles in drug efflux [36]. This protein is expressed in the liver, kidneys, and lungs [37]. Here, the patient carried the GG genotype of ABCB1 rs1128503 and presented with severe skin reactions. Preemptive genotyping of some ABCB1 SNPs, rs1128503 (c.1236A>G) in this case, can help to reduce the adverse reactions to chemotherapy drugs [38]. In colorectal cancer, capecitabine-treated patients with GG genotype exhibit a higher risk of neutropenia or hand-foot syndrome compared to those with AA genotype [39]. However, some studies on gefitinib have reported otherwise. A study of Chinese patients with NSCLC revealed that the GG genotype reduces the risk of diarrhea and skin rashes on gefitinib treatment [40]. In contrast, a systematic review with meta-analysis carried out by our group [41] demonstrated possible clinical implications of ABCB1 rs1128503 (c.1236A>G) in gefitinib-induced rash and diarrhea reactions. Therefore, the skin reactions observed in our patient may have been caused by this variant.
ABCG2 gene: we added the reference by Akasaka Keiichi et al. (2010; PMID 20035425), in the paragraph referring to this gene, and another study by Lina and Tang, (2018, PMID 29440914) reinforcing the hypothesis that heterozygous ABCG2 genes may not be reliable biomarkers for dermatological toxicity induced by gefitinib. We have reinforced the figure with the proposed mechanism for ADR, which was removed as suggested.
Rewritten the sixth paragraph of the discussion (3)
(lines 182-195)
ABCG2 is an important gene involved in the efflux of several drugs, including gefitinib [42]. Here, the patient was heterozygous for both ABCG2 rs223142 (421G>T) and rs2622604 (c.-20+614C>T). The consequences of these polymorphisms have not yet been fully elucidated, but one study showed that individuals carrying the GT genotype of the 421G>T (rs223142) polymorphism exhibit an increased risk of diarrhea compared to those with the GG genotype after gefitinib treatment [43]. However, a study with patients with NSCLC did not find associations with the T allele (heterozygous or homozygous) and gefitinib reactions [44]. The presence of adverse reactions, such as skin reactions or diarrhea, can be explained by the roles of transporters in the oral absorption and elimination pathways of gefitinib as ABCG2 is highly expressed in the intestine and liver [45]. A meta-analysis study that investigated the associations of ABCG2 genes and gefitinib toxicities showed that rs rs223142 (421G>T) may not be a reliable biomarker of dermatological toxicity induced by gefitinib [46]. Thus, the G variant of the ABCG2 421G>T polymorphism probably contributed to the adverse reactions observed in our patient.
We rewrote the conclusion section (4)
(lines 224-236)
Here, our findings showed a case of a patient with NSCLC, using gefitinib, doxycycline and metamizole, plus homozygosity in ABCB1 rs1128503 (c.1236A> G), heterozygosity in ABCG2 rs2231142 and rs2622604 (c.421G> T + c. -20 + 614T> C) and the presence of a non-functional component in the CYP3A5 variant and the presence of moderate to severe adverse reactions associated with the use of gefitinib.
There are several controversies about the contribution of these variants in the adverse reactions presented by the patient. However, we highlight the hypothesis that the ADRs induced by gefitinib may be caused by the non-functional variant CYP3A5*3 and in the homozygous variant of ABCB1 (rs1128503). We emphasize the importance of continuous study of pharmacogenetic analyses in predicting individual responses to treatment, particularly concerning adverse drug reactions. This is essential for advancing research on genetic variants as potential risk biomarkers.
- Abstract Lines 22-23 : For consistency, rs numbers of respective variants should be added.
Author's reply: RS numbers were added as suggested.
- Table 2 (page 5) : The column labeled ‘Patient Genotype/phenotype’ lists only genotypes for various genes. Remove the word ‘Phenotype’ from the heading of this column.
Author's reply: The requested changes were made.
- Figure 2 and related discussion is contrary to current documented evidence. Data presented in this manuscript does not support the assumptions on which this diagram is based. Therefore Figure 2 and related discussion should be removed entirely from this case report.
Author's reply: We agree with the reviewer that Figure 1 cannot be used to explain the adverse reaction observed. The figure has been removed as suggested.

Round 2
Reviewer 2 Report
Comments and Suggestions for Authors
I applaud the Authors' effort to revise the manuscript and agree with all edits except one. I am referring to the discussion about the involvement of the SNP rs223142 in the observed adverse reactions ( lines 182-195 of the revised draft of the manuscript). Here, the authors provide an excellent summary of the most recent state of research & understanding about the lack of any role of this SNP in observed adverse reactions, yet conclude the discussion by stating, ' Thus, the G variant of the ABCG2 421G>T polymorphism probably contributed to the adverse reactions observed in our patient. (Lines 194-195). Authors' inference, thus, is contrary to the results and should be modified to read , 'Thus the association of ABCG2 rs rs223142 in the current patient is questionable and warrants further investigation with additional cases'.
Author Response
Reviewer 2 (round2)
I applaud the Authors' effort to revise the manuscript and agree with all edits
except one. I am referring to the discussion about the involvement of the SNP
rs223142 in the observed adverse reactions ( lines 182-195 of the revised draft
of the manuscript). Here, the authors provide an excellent summary of the most
recent state of research & understanding about the lack of any role of this SNP
in observed adverse reactions, yet conclude the discussion by stating; Thus,
the G variant of the ABCG2 421G>T polymorphism probably contributed to the
adverse reactions observed in our patient. (Lines 194-195). Author' inference,
thus, is contrary to the results and should be modified to read , "Thus the
association of ABCG2 rs rs223142 in the current patient is questionable and
warrants further investigation with additional cases".
Author' reply: We apologize for the error and thank you for your observation. The lines have been corrected.
Rewritten fifth paragraph of the discussion (3) (lines 194 - 195)
"Thus, the association of ABCG2 rs223142 in the current patient is questionable
and warrants further investigation with additional cases".